# Impact of the Novel CoronaviruS (COVID-19) on Frontline PharmacIsts Roles and ServicEs: INSPIRE Worldwide Survey

**DOI:** 10.3390/pharmacy11020066

**Published:** 2023-03-29

**Authors:** Kaitlyn E. Watson, Dillon H. Lee, Mohammad B. Nusair, Yazid N. Al Hamarneh

**Affiliations:** 1EPICORE Centre, Department of Medicine, University of Alberta, Edmonton, AB T6G 2T9, Canada; kewatson@ualberta.ca; 2Faculty of Pharmacy and Pharmaceutical Sciences, University of Alberta, Edmonton, AB T6G 2H1, Canada; dillon1@ualberta.ca; 3Department of Sociobehavioral and Administrative Pharmacy, College of Pharmacy, Nova Southeastern University, Fort Lauderdale, FL 33314, USA; mnusair@nova.edu; 4Department of Clinical Pharmacy and Pharmacy Practice, College of Pharmacy, Yarmouk University, Irbid 211163, Jordan; 5Department of Pharmacology, Faculty of Medicine and Dentistry, University of Alberta, Edmonton, AB T6G 2H7, Canada

**Keywords:** pharmacists, COVID-19, pandemic, emergency response, pharmacy profession, roles, pharmacy services

## Abstract

Background: Pharmacy has been recognized as a vital healthcare profession during the COVID-19 pandemic. The primary objective of the INSPIRE Worldwide survey was to determine the impact of COVID-19 on pharmacy practice and pharmacists’ roles around the world. Methods: A cross-sectional online questionnaire with pharmacists who provided direct patient care during the pandemic. Participants were recruited through social media, with assistance from national and international pharmacy organizations between March 2021–May 2022. The questionnaire was divided into (1) demographics, (2) pharmacists’ roles, (3) communication strategies, and (4) practice challenges. The data were analyzed using SPSS 28, and descriptive statistics were used to report frequencies and percentages. Results: A total of 505 pharmacists practicing in 25 countries participated. The most common role that pharmacists undertook was responding to drug information requests (90%), followed by allaying patients’ fears and anxieties about COVID-19 (82.6%), and addressing misinformation about COVID-19 treatments and vaccinations (80.4%). The most common challenges were increased stress levels (84.7%), followed by medication shortages (73.8%), general supply shortages (71.8%), and inadequate staffing (69.2%). Conclusions: Pharmacists within this study were significantly impacted by the COVID-19 pandemic and took on new or adapted roles (e.g., providing COVID-specific information, managing patients’ emotions, and educating on public health measures) to meet the needs of their communities. Despite, the significant challenges (e.g., increased stress, supply chain challenges, addressing misinformation, and staffing shortages) faced by pharmacists, they continued to put their patients’ needs first and to provide pharmacy services.

## 1. Introduction

Pharmacists have been integral to the public health response to the ongoing COVID-19 pandemic and in relieving pressure on the overstretched healthcare system [1]. The International Pharmaceutical Federation (FIP) published interim guidelines for the pharmacy workforce outlining key activities to perform during the pandemic [2]. Such activities include: pharmacy operations and facilities, preventative measures, community pharmacy interventions and patient counseling, referral and isolation, pharmacy as an information resource, and more [2]. Furthermore, several other international papers have been published in the last couple of years to provide insight into the roles and responsibilities of pharmacists during COVID-19 pandemic [1,3,4,5,6]. An international scoping review of pharmacists’ roles during the first year of the COVID-19 pandemic found pharmacists were made visible in the roles and services they provided. They presented a conceptual framework model that categorized these visible roles into three categories-information, medication management, and public health [4]. Patients and community members were seeking out community pharmacists as a trusted and reliable source of information. Additionally, around the world, pharmacists were heavily relied upon to perform public health roles to increase the available healthcare resources and personnel (e.g., COVID-19 screening, testing, and public health messaging) [5,7,8,9,10,11]. A 2023 study exploring the lived experiences of community pharmacists expanded this conceptual framework model and found leadership was another role category that was made visible by the pandemic [12].

The pharmacy profession, like so many other avenues of the healthcare system, was not prepared for the impact of the COVID-19 pandemic and the prolonged need to adapt their response. The demands for pharmacy services were beyond the expectations of pharmacists and in some jurisdictions beyond their scope of practice. Such demands forced them to juggle the need for pandemic prevention and control with their usual clinical activities and services [13].

This is not the first-time pharmacists have been relied on during a crisis. One of the best lessons learned from the H1N1 influenza virus pandemic response was that pharmacists are ideally placed to provide public health services and education, assist with vaccination, and advance public health response capability [14,15]. Furthermore, a scripted surge pharmacy pandemic exercise conducted by the Centers for Disease Control and Prevention and the National Association of County and City Health Officials highlighted that pharmacies would likely improve access to vaccinations and medications during a future pandemic [16]. Indeed, it has been reported that pharmacies improved community emergency response by collaborating with local health departments during the influenza outbreak [17]. However, this was not translated into actionable disaster plans and thus the lesson was repeated during the COVID-19 pandemic. This not unique to pandemics either, with previous disaster events have identified the importance of pharmacists in disaster response and recovery and calls for better integration of pharmacists into disaster planning and emergency management. In 2019, an international expert panel came to consensus on 43 pharmacists’ roles across the disaster management cycle-prevention/mitigation, preparedness, response, and recovery [18].

Pharmacy has been recognized as a vital healthcare profession during the COVID-19 pandemic. Impact of the Novel CoronaviruS (COVID-19) on Frontline Pharmacist Roles and ServicEs in Canada (the INSPIRE questionnaire [19]) identified the impact of the pandemic on Canadian pharmacists’ roles and pharmacy services. The INSPIRE questionnaire was designed in April 2020 and collected data online from 27 May to 31 July 2020. The INSPIRE questionnaire showed that the pandemic has significantly impacted the roles that Canadian pharmacists were required to undertake and the services they provided to their patients in the early phase of the pandemic. Since the COVID-19 pandemic is still ongoing and new viral strains are emerging, an update on its impact is warranted. Additionally, a comprehensive snapshot of that impact on international pharmacy practice would be beneficial. Thus, the primary objective of the INSPIRE Worldwide questionnaire was to determine the impact of the COVID-19 pandemic on pharmacy practice and pharmacists’ roles and services around the world. 

## 2. Methods

### 2.1. Study Design

This study was a descriptive exploratory, nonexperimental cross-sectional questionnaire. This manuscript was prepared based on the STROBE checklist [20].

### 2.2. Participants

Participants were frontline pharmacists around the world who have been impacted by the COVID-19 pandemic.

### 2.3. Participant Recruitment

Pharmacists were recruited through national and international pharmacy advocacy organizations (e.g., Pharmaceutical Society of Australia, Canadian Pharmacists Association, Alberta Pharmacists Association, International Pharmacy Federation, American Health-Systems Pharmacists Association, Jordanian Pharmacists Association), social media, word of mouth, and snowball sampling. The questionnaire was provided in English and French. Further translations were not possible at the time of study conduct.

The respondents had the opportunity to review the study information sheet. Once they reviewed the information sheet and decided to take part, they were asked to provide their consent. Once consent was obtained, participants were able to access the questionnaire. Pharmacists were eligible to take part if they provided direct patient care during the COVID-19 pandemic. The online questionnaire was available from March 2021–May 2022. 

### 2.4. Data Collection Instrument

The questionnaire was previously developed, piloted, and published in the INSPIRE Canada study [19]. The questionnaire was divided into 4 sections: (1) demographics, (2) COVID-19 pharmacists’ roles and services, (3) communication strategies, and (4) specific COVID-19 challenges experienced. The questions were updated using the available literature and knowledge of emerging challenges faced by the pharmacy profession internationally, and new challenges identified since the original study was conducted. The updated questionnaire was pilot tested with 10 national and international pharmacists before distribution. This was to ensure the questions were clear and updated appropriately to reflect the current stage of the pandemic.

The online questionnaire (Appendix A) was developed and distributed using the secure electronic data capture web application Research Electronic Data Capture (RedCap).

### 2.5. Data Analysis

The data were exported and analyzed using the Statistical Package for the Social Sciences (SPSS), version 28. Descriptive statistics were used to provide the frequencies and percentages of the impact of the COVID-19 pandemic on pharmacists’ roles and services. Pharmacists were not required to answer every question; thus, the denominator may differ for each question depending on the number of pharmacists who opted to respond. This study provides a snapshot of the impact of the COVID-19 pandemic on frontline pharmacists, further statistical analysis was not within the scope of the study.

Ethics approval was obtained from the University of Alberta Health Research Ethics Board, approval number Pro00108598. 

## 3. Results

A total of 505 pharmacists practicing in 25 countries provided consent and completed the questionnaire. Table 1 provides a description of the participants’ characteristics. The majority of the respondents were from the regions of the Americas (70.7%), followed by the Eastern Mediterranean Region (15.6%) then the Western Pacific region (8.1%) (Table 1).

### 3.1. Pharmacists’ Roles and Services

Table 2 outlines the pharmacist roles and services performed by the participants during the COVID-19 pandemic. The most common role that participants undertook was responding to drug information requests (89.4%), followed by allaying patients fears and anxiety (82.8%), and educating people on public health measures (81.3%).

The roles and services differed according to the practice settings and reflected everyday practice for pharmacists. For example, the pharmacists’ role of delivering medications was applicable to 83.4% of the community pharmacist participants and only 17.6% for those practising in acute care. 

Interestingly, when looking at the WHO regions, the pharmacists’ role of administering the COVID vaccine differed. Participants from the region of the Americas reported the highest percentage (51.4%), followed by the Eastern Mediterranean region (32.9%), the European region (25%), and Western Pacific region (9.8%) (Appendix A). Compounding hand sanitizers was highest for the Eastern Mediterranean and European regions (48.1% and 41.7%, respectively) compared to only 26.8% for the Western Pacific region and 7.3% for the region of the Americas. All regions reported they had pharmacists performing the role of reporting domestic violence, with the Eastern Mediterranean region being the highest at 21.5% of participants. 

### 3.2. Challenges

Most participants (82.2%) reported that increased stress level was their biggest challenge in responding to the COVID-19 pandemic (Table 3). This was followed by challenges with the supply chain for medicines (71.3%), and managing misinformation (64.4%). Overall, these challenges had the least impact on the Eastern Mediterranean region participants’ practice. The following challenges were more commonly reported by participants from the Western Pacific Region: decreased supply of medications, prescription surge, general supply shortage, concern for safety, extra cost associated with following the public health regulations, and loss of business. Time required to develop safety plans for COVID-19, lack of access to COVID-19 vaccines, and insurance issues had higher impact on participants from the region of the Americas (Table 3).

### 3.3. Engaged with Disaster Response

Only 26.4% of the total participants were engaged with local disaster and public health agencies to coordinate their pharmacy response to the COVID-19 pandemic. Additionally, 35.8% were consulted about the pandemic in terms of pharmacist roles and expectations of pharmacy services (Table 4). Of those that were consulted, 43% were consulted by pharmacy organizations, 37.4% were consulted by administration at their hospital, 31.8% were consulted by public health services, and 24.6% were consulted by both corporate and government, respectively. Across the different WHO regions, pharmacy organizations were the most common group that consulted the participating frontline pharmacists. Very few of the participants were consulted by local disaster response organizations. 

## 4. Discussion

This study provides an international overview of pharmacists’ roles and services and the challenges they faced in their response to the ongoing COVID-19 pandemic. There were differences in pharmacists’ roles or services observed across the different practice settings and WHO regions as anticipated. Not all pharmacists’ roles and services discussed in this study are applicable to every practice setting or within the scope of practice of every jurisdiction. Literature has identified the essential role pharmacists provided in administering the COVID vaccine to increase scale and accessibility for mass prophylaxis [21,22,23]. Despite this, fewer than half of the participants stated they performed this role. Our study highlights the importance of pharmacists’ information role, which was evident with most participants responding to drug information requests, allaying patients’ fears and anxiety, and educating people on public health measures. An international conceptual framework model of pharmacists’ role changes during the COVID-19 pandemic and the current study contribute to better understanding of the shift in the visibility of pharmacists’ information role [4]. The international scoping review suggested pharmacists pre-COVID were only seen by their colleagues and patients as ‘drug experts’ but as the pandemic has progressed, pharmacists have been identified in society as a key reliable and accessible source of public health information [4]. This is in line with our findings of the demand for pharmacists’ information roles. Additionally, other reviews conducted about pharmacists’ roles during the COVID-19 pandemic found similar results [24,25]. This study also provides an extension to the early snapshot of the impact of the pandemic on Canadian pharmacists presented in the original INSPIRE Questionnaire [12]. The initial INSPIRE questionnaire, which was conducted in early 2020 with Canadian pharmacists, found that the most common pharmacists’ roles and challenges were related to supplies and shortages [19]. This included rationing supplies of medications for equitable access, managing personal protective equipment supplies, and other general supply challenges arising from interruptions to the global supply chains. The biggest challenge faced by participants in this study and in the original INSPIRE questionnaire was the increased stress level. Eighteen months later and the challenge of medicine shortages continue to be a major issue. The challenges of the supply chain were not unique to the pharmacy profession, with a recent review highlighting the disruptions faced by the supply chain [26].

The global pandemic placed tremendous pressures on all levels of the healthcare system. Pharmacists consequently experienced increased demands and reduced resources in their practice environments [13,27,28]. An Australian study of pharmacists’ burnout due to COVID-19 found that there were high rates of emotional exhaustion and depersonalization [29]. The same others used the job demands-resource framework to describe the pharmacists’ experiences [27]. The job demands-resource framework theorizes that for an individual to cope with high demands at a workplace, they require sufficient resources, which include fiscal and physical resources, as well as emotional and professional assets, like feedback and support [27,28]. Since burnout commonly occurs when there is a sustained mismatch between demands and resources, the frequent reports of increased stress from this questionnaire certainly raise an alarm and match findings from other studies on the impact of the COVID-19 pandemic on pharmacists and the reported high rates of pharmacists’ burnout. Thus, it is critical to identify ways to support their psychological wellbeing [29]. The job demands-resources model explains that increasing resources during the times of challenges, such as providing additional management and peer supports may reduce the workload by increasing efficiency. Similarly, a study from the perspective of Chinese pharmacists recommended that staff be provided adequate training to perform their information, public health, and medication management roles [13].

Interestingly, only 26.4% of the total participated reported that they were engaged in local public health and emergency response. While it is clear that pharmacists’ efforts around the world have been extraordinary in responding to and recovering from the COVID-19 pandemic, they still need to be linked to the existing emergency and public health coordinated response. The contributions of the global pharmacy profession need to be well documented and integrated in future disaster and business continuity plans. This is highlighted in the FIP 2006 statement on “The Role Of The Pharmacist In Crisis Management: Including Manmade And Natural Disasters And Pandemics” which recommends governments and organizations consider and include pharmacists when planning for disasters and emergencies [30,31].

## 5. Limitations

This study provides an international snapshot of the challenges faced by pharmacists working on the frontlines during the COVID-19 pandemic and the roles they performed. We did not power this study to provide a breakdown comparison across regions or countries, and thus further statistical analysis was outside of the scope of this study. Additionally, the survey was available in English and French, but due to the limitations of the research, it was not possible to translate the survey into other languages, which may affect the generalisability of the results. While this may have impacted the results, there was good representation across five of the WHO regions. Further research could explore the implications of the impact of the COVID-19 pandemic to the business of pharmacy. Our sampling methodology, through national and international pharmacy advocacy organizations and working groups, social media, and word of mouth, could have impacted the representation of the international pharmacy profession. Despite that, we had representation from five of the WHO regions and 25 countries.

## 6. Conclusions

Pharmacists within this study were significantly impacted by the COVID-19 pandemic and took on new or adapted roles (e.g., providing COVID-specific information, managing patients’ emotions, and educating on public health measures) to meet the needs of their communities. Despite the significant challenges (e.g., increased stress, supply chain challenges, addressing misinformation, and staffing shortages) faced by pharmacists, they continued to put their patients’ needs first and to provide pharmacy services. This study is a snapshot into the impact of the COVID-19 pandemic on international frontline pharmacists. It informs healthcare systems and health policy about the challenges faced by the pharmacy profession which can be the launching platform for future disaster planning for public health crises. 

## Figures and Tables

**Table 1 pharmacy-11-00066-t001:** Participant Characteristics.

	Number of Participants (*n* = 505)	Percentage (%)
**Practice Setting**		
Community Pharmacy (e.g., independent, chain, franchise)	291	57.6
Team-based Community Practice (e.g., primary care network, family health team)	25	5.0
Acute Care/Inpatient Care (e.g., hospital)	125	24.8
Ambulatory/Outpatient Clinic (e.g., specialty clinics)	44	8.7
Continuing or long-Term Care	10	2.0
Worked across multiple different practice settings	87	17.2
Other (e.g., academia, government, research, industry, etc.)	20	4.0
**WHO Region ***		
African Region	2	0.4
Eastern Mediterranean Region	79	15.6
European Region	12	2.4
Region of the Americas	357	70.7
Western Pacific Region	41	8.1
Undisclosed	14	2.8

* Participants countries were assigned according to the WHO Regions, the categories can be found at this link https://www.who.int/countries (accessed on 13 January 2023).

**Table 2 pharmacy-11-00066-t002:** Pharmacists’ roles and services during the COVID-19 pandemic.

	Yes	No	Role not Relevant to Participant’s Practice Setting	Role not Relevant to Participant’s Country
Responding to drug information requests (*n* = 501)	448 (89.4%)	33 (6.6%)	20 (4.0%)	0
Allaying patients fears and anxiety about COVID-19 (*n* = 499)	413 (82.8%)	64 (12.8%)	20 (4.0%)	2 (0.4%)
Educating the public on reducing the spread of COVID-19 (e.g., handwashing, physical/social-distancing) (*n* = 503)	409 (81.3%)	66 (13.1%)	27 (5.4%)	1 (0.2%)
Addressing misinformation on COVID-19 treatments and vaccination to patients (*n* = 502)	397 (79.1%)	74 (14.7%)	30 (6.0%)	1 (0.2%)
Performing medication reviews (*n* = 498)	391 (78.5%)	74 (14.9%)	32 (6.4%)	1 (0.2%)
Managing and/or monitoring patients’ chronic diseases (*n* = 501)	362 (72.3%)	81 (16.2%)	58 (11.6%)	0
Rationing medicine supplies (*n* = 501)	341 (68.1%)	107 (21.4%)	51 (10.2%)	2 (0.4%)
Treating ambulatory conditions (e.g., mild constipation, mild pain) (*n* = 501)	330 (65.9%)	92 (18.4%)	78 (15.6%)	1 (0.2%)
Renewing/Extending prescriptions (*n* = 502)	293 (58.4%)	108 (21.5%)	93 (18.5%)	8 (1.6%)
Administering influenza and other vaccines (*n* = 502)	278 (55.4%)	143 (28.5%)	68 (13.6%)	13 (2.6%)
Delivering medications to patients at home (*n* = 501)	272 (54.3%)	128 (25.6%)	97 (19.4%)	4 (0.8%)
Providing PPE supplies to patients (e.g., face masks) (*n* = 500)	258 (51.6%)	192 (38.4%)	49 (9.8%)	1 (0.2%)
Prescribing emergency supply refills (*n* = 501)	255 (50.9%)	130 (26.0%)	110 (22.0%)	6 (1.2%)
Administering the COVID-19 vaccine (*n* = 501)	225 (44.9%)	197 (39.3%)	66 (13.2%)	13 (2.6%)
Providing telehealth or tele-pharmacy consults (*n* = 501)	226 (45.1%)	201 (40.1%)	66 (13.2%)	8 (1.6%)
Addressing misinformation on COVID-19 treatments and vaccination to other healthcare providers (*n* = 503)	224 (44.5%)	207 (41.2%)	69 (13.7)	3 (0.6%)
Guiding policy development on COVID-19 (*n* = 500)	139 (27.8%)	284 (56.8%)	73 (14.6%)	4 (0.8%)
Advocating for COVID-19 public health messaging (e.g., radio, television, community outreach) (*n* = 499)	127 (25.5%)	290 (58.1%)	79 (15.8%)	3 (0.6%)
Participating on COVID-19 taskforce (*n* = 498)	127 (25.5%)	299 (60.0%)	69 (13.9%)	3 (0.6%)
Providing drive-thru pharmacy services (e.g., anticoagulant clinic, INR testing, medication dispensing, COVID-19 testing) (*n* = 499)	122 (24.5%)	259 (51.9%)	107 (21.4%)	11 (2.2%)
Providing Psychological First Aid (e.g., identifying patients at risk of mental health crises) (*n* = 500)	106 (21.2%)	315 (63.0%)	72 (14.4%)	7 (11.4%)
Compounding hand sanitizers (*n* = 500)	84 (16.8%)	318 (63.6%)	93 (18.6%)	5 (1.0%)
Contributing to/establishing a field hospital for COVID-19 (*n* = 499)	55 (11.0%)	320 (64.1%)	117 (23.5%)	7 (23.5%)
Coordinating clinical trial management specific to COVID-19 treatments (*n* = 499)	51 (10.2%)	334 (66.9%)	107 (21.4%)	7(1.4%)
Reporting domestic violence (*n* = 499)	43 (8.6%)	378 (75.8%)	72 (14.4%)	6 (1.2%)
Working in a field hospital for COVID-19 (*n* = 500)	28 (5.6%)	345 (69.0%)	119 (23.8%)	8 (1.6%)

**Table 3 pharmacy-11-00066-t003:** Challenges that impacted pharmacists’ practice during the COVID-19 pandemic.

	Eastern Mediterranean Region (*n* = 79)	European Region (*n* = 12)	Region of the Americas (*n* = 357)	Western Pacific Region (*n* = 41)	African Region (*n* = 2)	Undisclosed Location (*n* = 14)	Total (*n* = 505)
Increased stress level	46 (58.2)	10 (83.3)	319 (89.4)	34 (82.9)	0	6 (42.9)	415 (82.2)
Decreased supply of medications	30 (38.0)	8 (66.7)	279 (78.2)	36 (87.8)	2 (100)	5 (35.7)	360 (71.3)
Correcting misinformation with patients	26 (32.9)	8 (66.7)	258 (72.3)	26 (63.4)	1 (50)	6 (42.9)	325 (64.4)
Inadequate staffing	31 (39.2)	4 (33.3)	257 (72.0)	25 (61.0)	2 (100)	3 (21.4)	322 (63.8)
General supply shortage	25 (31.7)	7 (58.3)	244 (68.4)	36 (87.8)	2 (100)	7 (50.0)	321 (63.6)
Concern for safety	27 (34.2)	6 (50.0)	245 (68.6)	31 (75.6)	1 (50)	5 (35.7)	315 (62.4)
Lack of personal-protective equipment	39 (49.4)	7 (58.3)	201 (56.3)	28 (68.3)	1 (50.0)	6 (42.9)	282 (55.8)
Unfair patient expectations	28 (35.4)	7 (58.3)	202 (56.6)	24 (58.5)	0	5 (35.7)	266 (52.7)
Lack of time for clinical counseling	30 (38.0)	6 (50.0)	199 (55.7)	18 (43.9)	1 (50.0)	4 (28.6)	258 (51.1)
HR-related issues	30 (38.0)	8 (66.7)	194 (54.3)	19 (46.3)	1 (50.0)	3 (21.4)	255 (50.5)
Inadequate time for breaks/meals	20 (25.3)	4 (33.3)	199 (55.7)	20 (48.8)	1 (50.0)	4 (28.6)	248 (49.1)
Prescription surge	11 (13.9)	4 (33.3)	168 (47.1)	26 (63.4)	1 (50.0)	4 (28.6)	214 (42.4)
Time required to develop safety plans for COVID-19	26 (32.9)	3 (25.0)	151 (42.3)	12 (29.3)	0	2 (14.3)	194 (38.4)
Lack of access to COVID-19 vaccines	15 (19.0)	1 (8.3)	162 (45.4)	6 (14.6)	0	3 (21.4)	187 (37.0)
Insurance issues	19 (24.1)	1 (8.3)	141 (39.5)	5 (12.2)	0	5 (35.7)	171 (33.9)
Extra cost associated with following the public health regulations	16 (20.3)	3 (25.0)	120 (33.6)	22 (53.7)	0	2 (14.3)	163 (32.3)
Feeling unsure of pharmacy’s role or responsibilities	20 (25.3)	5 (41.7)	114 (31.9)	13 (31.7)	0	2 (14.3)	154 (30.5)
Loss of business	15 (19.0)	4 (33.3)	87 (24.4)	17 (41.5)	1 (50.0)	2 (14.3)	126 (25.0)
Childcare	16 (20.3)	1 (8.3)	95 (26.6)	10 (24.4)	0	3 (21.4)	125 (24.8)
Not feeling confident about delivering expanded scope of practice services during COVID-19	9 (11.4)	2 (16.7)	82 (23.0)	5 (12.2)	0	1 (7.1)	99 (19.6)

**Table 4 pharmacy-11-00066-t004:** A comparison by WHO Regions of whether participants were consulted about the COVID-19 pandemic by organizations.

	Eastern Mediterranean Region (*n* = 79)	European Region (*n* = 12)	Region of the Americas (*n* = 357)	Western Pacific Region (*n* = 41)	African Region (*n* = 2)	Undisclosed Location (*n* = 14)	Total (*n* = 505)
** *Were you consulted about COVID-19 in terms of pharmacist roles/expectations for pharmacy services?* **							
Yes	44(55.7)	7(58.3)	108(30.5)	13(31.7)	0	7(58.3)	179(35.8)
** *If yes, by whom?* **							
Pharmacy organizations	19 (43.2)	5 (71.4)	40 (37.0)	10 (76.9)	0	2 (28.6)	77 (43.0)
Administration teams in hospitals	6 (13.6)	3 (42.9)	54 (50.0)	3 (23.1)		3 (42.9)	67 (37.4)
Public health services	17 (38.6)	1 (14.3)	29 (26.9)	7 (53.9)		1 (14.3)	57 (31.8)
Government	12 (27.3)	1 (14.3)	24 (22.0)	5 (38.5)		3 (42.9)	44 (24.6)
Corporate	12 (27.3)	1 (14.3)	25 (23.2)	5 (38.5)		1 (14.3)	44 (24.6)
Local disaster response organizations	5 (11.4)	0 (0.0)	8 (7.4)	3 (23.1)		1 (14.3)	17 (9.5)
Other (e.g., university, schools, colleagues)	2 (4.6)	1 (14.3)	6 (5.6)	0 (0.0)		0 (0.0)	9 (5.0)

## Data Availability

The data is available in the Appendix A.

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
