# Peer review of "Impact of the Novel CoronaviruS (COVID-19) on Frontline PharmacIsts Roles and ServicEs: INSPIRE Worldwide Survey"

_pharmacy, 2023, doi:10.3390/pharmacy11020066_

Round 1

Reviewer 1 Report

Paper pharmacy-2244795 “Impact of the Novel Coronavirus (COVID-19) on Frontline Pharmacists Roles and Services: INSPIRE Worldwide Survey”

Comments

This study investigates the impact of the novel coronavirus (COVID-19) on frontline pharmacists roles and services. I think the paper fits well the scope of the journal and addresses an important subject. However, a number of revisions are required before the paper can be considered for publication. There are some weak points that have to be strengthened. Below please find more specific comments:

*Abstract: The abstract seems to be adequate. No comments.

*Keywords: The keywords seem to be reasonable.

*Relevant literature: The authors discuss a number of studies dealing with the impacts of COVID-19 but many relevant studies highlighting the negative impacts of COVID-19 on the normal operations in different domains (e.g., closure of various facilities due to COVID-19 lockdowns, supply chain disruptions, transition to remote operations, low productivity of employees due to high rick perception to COVID, etc.) are missing. Please double check and acknowledge the relevant studies, including but not limited to the following:

The Impact of trust and risk perception on the acceptance of measures to reduce COVID‐19 cases. Risk Analysis 2021, 41(5), pp.787-800.

Supply chain disruption during the COVID-19 pandemic: Recognizing potential disruption management strategies. International Journal of Disaster Risk Reduction 2022, 75, p.102983.

An Assessment of Social Distancing Obedience Behavior during the COVID-19 Post-Epidemic Period in China: A Cross-Sectional Survey. Sustainability 2021, 13(14), p.8091.

Implementing Public Health Strategies—The Need for Educational Initiatives: A Systematic Review. International Journal of Environmental Research and Public Health 2021, 18(11), p.5888.

Pandemic supply chain research: A structured literature review and bibliometric network analysis. Logistics 2021, 5(1), p.7.

Supply chains under COVID-19 disruptions: literature review and research agenda. In Supply Chain Forum: An International Journal, 2022, (Vol. 23, No. 1, pp. 81-95). Taylor & Francis.

*Literature review: Please double check for the most recent and relevant studies published over the last 2-3 years.

*More information should be provided regarding the actual data used in this manuscript. Did you experience any challenges at the data collection stage? This aspect should be discussed more in detail throughout the manuscript.

*The manuscript could benefit from additional figures and tables to make the presentation of this work more thorough. More detailed discussions are needed to make sure that the readers will have a deep understanding of the major outcomes from this research.

*The discussions and coverage of managerial insights seem to be rather short. Please try to expand those.

*The conclusions section should expand on limitations of this study and future research needs. I suggest listing the bullet points.

Reviewer 2 Report

This is an interesting paper on a timely topic. The article presents the important issue of the expanding role of a pharmacists during the Covid-19 pandemic.  Nevertheless, the manuscript requires a minor revision in order to make it valuable material for publication.

 Introduction

The Introduction  is quite short, and mainly focused on the situation of the pharmacists and their role during the Covid-19 pandemic in Canada. It is well known  that the increase in the scope of the role of the pharmacist and its importance in the health care system during the pandemic was noticeable in most countries, which is confirmed by  extensive literature.  Therefore, the authors should refer to the situation in other countries, what the title of the manuscript would indicate.  The references might be extended to a larger number of literature items. It will increase the value of the introduction as well as the entire study.

 Methods

Description of the sampling methodology applied to the study, indicates that the sample is not representative. Therefore, the calculation of the statistical error on the basis of a sample selected in the way presented by the authors (social media and professional organizations) is considered to be unfounded. This mechanism is only used for a representative sample.

Discussion

It is necessary to supplement the discussion with international literature raising the issue of the role of a pharmacist during the pandemic from the global perspective.

Conclusions

They seem too general , sounding like a truism . Perhaps they might be specified in relation to the results.

Reviewer 3 Report

Comments:

The authors (Kaitlyn et al.) have reported on the impact of the novel coronavirus (COVID-19) on frontline pharmacists' roles and services through the INSPIRE Worldwide Survey. While this work is new, there are several reports available in the literature. Overall, the manuscript is suitable for publication, but there are some comments that can help the author in future submissions:

1.      The introduction needs improvement to provide a clear and comprehensive overview of the topic while ensuring that it is well-supported and includes the most recent updates.

2.      The author should include the demographics survey.

3.      The use of "and" in lines 39-42 in the introduction should be reduced.

4.      It would be useful to know more about the mental health of both pharmacists and patients during the pandemic.

5.      In the supporting data, the author provides a descriptive analysis of INSPIRE worldwide by WHO region (N = 505), but the sub-heading is missing in the first section. This section should be explained.

6.      The author explains the pharmacist roles and services, but it is not clear whether the characteristics of the study participants are divided by gender. The author should include this information to better understand patients' fears and anxiety towards COVID-19, which directly affects the role of pharmacists.

Round 2

Reviewer 1 Report

The authors took seriously my previous comments and made the required revisions in the manuscript. The quality and presentation of the manuscript have been improved. Therefore, I recommend acceptance.

Author Response

Thank you for reviewing our paper.